# The Fate of Chlorophylls in Alkali-Treated Green Table Olives: A Review

**DOI:** 10.3390/molecules28186673

**Published:** 2023-09-18

**Authors:** Roberto Ambra, Gianni Pastore, Fausta Natella

**Affiliations:** CREA (Council for Agricultural Research and Economics), Research Centre for Food and Nutrition, 00178 Rome, Italy; roberto.ambra@crea.gov.it (R.A.); giovanni.pastore@crea.gov.it (G.P.)

**Keywords:** table olives, alkali processing, chlorophylls derivatives, color adulteration

## Abstract

This paper reviews the current knowledge regarding modifications to chlorophylls during the processing of green table olives treated with alkali. Particular attention is given to the pheophytinization reactions (substitution of Mg^2+^ by 2H^+^ in the chlorophyll chromophore group) that can take place because of pH and/or temperature changes and the possible sequential substitution of the 2H^+^ with Cu^2+^ within the chlorophyll porphyrin ring. These reactions may have a direct impact on the commercial value of olive productions as some naturally forming Cu–chlorophylls complexes (i) are identical to strictly forbidden colorants for table olives (E141) and (ii) have been identified as responsible for the unwelcome appearance of the so-called green staining alteration (characterized by bluish-green zones distributed over the olive skin of the drupes).

## 1. Introduction

*Olea europeae* is one of the oldest cultivated trees, domesticated 6000 years ago in the Mediterranean basin and now grown commercially in almost all countries between latitudes of 30° and 45° [1]. Table olives represent almost 10% of all harvested olives in the world, with total worldwide production almost reaching 3 million tons in 2022 and a global consumption that has nearly doubled in the last 30 years. Table olives are a valuable source of energy and phytochemicals with health benefits.

Four main approaches are currently employed for the processing of raw olives into table olives, namely, spontaneous fermentation without any chemical debittering treatment (as in Greek-style olives), immersion in alkali lye followed by fermentation (as in Spanish-style green olives), immersion in alkali lye and salt without fermentation (the Castelvetrano method), and immersion in lye followed by oxidization (as with California-style black ripe olives) [2]. With respect to alkali debittering approaches, the subject of this review, in the Spanish (or Sevillian) method, olives are kept for few hours in a NaOH solution to hydrolyze oleuropein, then washed to remove alkali and conserved 1–4 months in a NaCl solution for a mild lactic fermentation. In the faster debittering process using the Castelvetrano (or Campo Real) method, after a few hours in a NaOH solution, drupes are kept for up to one month in a commingling lye/NaCl brine, forgoing the posterior lactic acid fermentation process. Such conditions allow for rapid processing while maintaining a bright green color. 

The color of Spanish and Castelvetrano-style olives we buy at the supermarket is the result of chemical and enzymatic transformations affecting the olive’s natural dyes, including chlorophylls. For example, pH changes associated with treatments necessary for making olives edible induce the hydrolysis of bitter molecules [3] but promote chlorophyll pheophytinization reactions. Chemically, pheophytinization reactions consist of the replacement of the central magnesium atom of the tetrapyrrole by two hydrogen atoms [4]. Reactions are favored by acidic and mild heat conditions. Because of the loss of Mg^2+^ from the porphyrin ring, the bright green color of the chlorophyll turns to olive brown. Other important chemical reactions can influence the drupes’ color in an opposite fashion and thus are referred to as regreening. Such phenomena are associated with the insertion of endogenous or exogenous metals in the olives (Zn^2+^ or Cu^2+^) into the porphyrin ring of chlorophylls. Such complexes possess greater chemical stability in acid compared to the original compounds thanks to the higher metal–porphyrin bond energies [5]. Regreening phenomena are exploited by the food industry to preserve the desirable fresh-like green color of vegetables [5,6]. On the other hand, Cu–chlorophyll complexes can also cause an alteration called green staining [7,8,9,10], characterized by bluish-green zones randomly distributed over the olive skin and affecting the “natural” olive color. Notably, these alterations have been mainly reported in Gordal variety debittered through the Castelvetrano method. Moreover, these naturally formed Cu–chlorophyll complexes can bring incorrect fraudulence assessments since some of these molecules are contained in artificial dyes (E-141, which is marketed according to its solubility—“i” for the liposoluble forms or “ii” for the hydrophilic ones) that are actually banned for olive usage both by the US Food and Drug Administration and by the European Union [11]. Color changes in Spanish and Castelvetrano-style olives are of great concern for olive factories as color is very important for consumer acceptance [12]. This paper is aims to review the existing knowledge on such matters. 

## 2. Color Changes in Olive Drupes during Ripening

The final green color of fresh olives is due to the color combination of chlorophylls (a and b) with carotenoids. The main green pigment is chlorophyll a, followed by chlorophyll b. Yellowish carotenoids include lutein, the major pigment, followed by β-carotene and, in lower proportions, neoxanthin and violaxanthin, as well as antheraxanthin in much smaller amounts [13]. Except for antheraxanthin, which is absent in very ripe fruits, the chlorophyllic and carotenoid composition pattern of the fruits does not change qualitatively during ripening. However, both chlorophylls and carotenoids decrease during ripening, with the former having a faster degradation rate [13,14], causing a color shift from intense to light and yellowish-green. Regarding the degradation rate of the two chlorophylls, contradictory results have been reported. While Knee observed an equivalent degree of degradation for both chlorophylls [15], Gross and Shimokawa independently reported the preferential degradation of chlorophyll a [16] and b [17], respectively. Subsequently, Minguez-Mosquera and colleagues noticed that the degradation rate is somehow dependent on the maturation stage; it is higher for chlorophyll b when olives are green or ripe and higher for chlorophyll a in the intermediate stage [13]. More recent data show that the degradation pattern is also influenced by cultivar-specific characteristics [14]. Finally, subtle and local changes in chlorophyll degradation rates occurring during ripening can be responsible for reddish spots on the skin of olives [13].

Despite the decline in chlorophyll and carotenoid levels, other coloration changes occur during ripening as a consequence of the synthesis of several anthocyanic compounds. The most abundant anthocyanins present in ripe fruits are cyanidin 3-O-rutinoside and cyanidin 3-O-glucoside, followed by minor amounts of delphinidin-3-rhamnoglucoside, peonodin-3-glucoside acylated with p-coumaric acid, cyanidin-3-glucoside acylated with p-coumaric acid, and cyanidin-3-O-rutinoside acylated with caffeic acid [18]. The hydrophilic characteristics of these anthocyanic compounds determine the red/violet coloration in drupes, which is not only limited to the skin of the drupes, as it also affects the fruit internally [13].

## 3. Pigment Changes during Alkali Olive Processing

Two main alkali debittering approaches are normally adopted to make green olives edible. In the Spanish method, olives are treated with 2–5% lye for 6–10 h, washed to remove excess lye, and then placed in an 8–10% salt brine at around 25 °C, where they undergo fermentation for up to four months. In the faster debittering process using the Castelvetrano method, which is very popular in Italy and employed mainly for olives from autochthonous Sicilian cultivars, the fermentation step is omitted and, depending on drupe ripeness, debittering starts in a 2–3% lye solution in plastic barrels for a few hours, followed by the addition of salt (3–4 kg/100 L) and immersion in a commingling lye/NaCl brine for a deeper penetration of NaOH into the drupes (usually up to the pit) for 2–4 weeks, after which the olives are ready for consumption, following the removal of excess lye.

In general, during processing, the degradation of the olives’ chlorophylls follows two main pathways [19]. The first one is driven by alkaline conditions that can activate chlorophyllase enzymes or provoke oxidative reactions that affect the chlorophyll isocyclic ring, producing allomerized chlorophylls with chlorin-type (series *a*) and rhodin-type (series *b*) structures [20]; the other one is a chemical modification promoted by the acidic environment generated by the fermentation process. The chlorophyllases catalyze the de-esterification of the phytol ester, a reaction that does not affect the chromophoric properties of the resulting chlorophyllides. Conversely, the acidic fermentation medium is responsible for the drastic color change due to the replacement of Mg^2+^ by 2H^+^ in the porphyrin ring of chlorophylls and chlorophyllides, which are converted to pheophytins and pheophorbides, respectively. This reaction, known as pheophytinization, is responsible for the color switch from green to greyish-brown [21]. Figure 1 summarizes the chemical reactions that occur during the degradation of the chlorophylls of olives, while Figure 2 depicts the molecular structures of chlorophylls and their derivatives and lists the references (in the legend) that identified each single molecule in table olives for the first time.

In particular, during the allomerization reaction, the isocyclic ring of the chlorophyll structure is oxidized by the triplet molecular oxygen (^3^O_2_) at the level of the C-13^2^ residue [22]. Oxidized allomerized derivatives are also products of natural chlorophyll turnover and catabolism [23]; however, the presence of NaOH during olive processing strongly enhances the process, yielding a higher amount of 13^2^-OH- and 15^1^-OH-lactone-chlorophyll allomerized derivatives [22,24]. Following solvolysis reactions, allomerized derivatives characterized by an open isocyclic ring (chlorin- and rhodin-type structures, for series a and b, respectively) are also formed (Figure 1 and Figure 2). Other modified complexes include 15^2^-Me-phytol-chlorin *e*_6_ ester, 15^2^-Me-phytol-rhodin *g*_7_ ester, and 15^2^-Me-phytol-isochlorin *e*_4_ ester [22,25].

**Figure 2 molecules-28-06673-f002:**
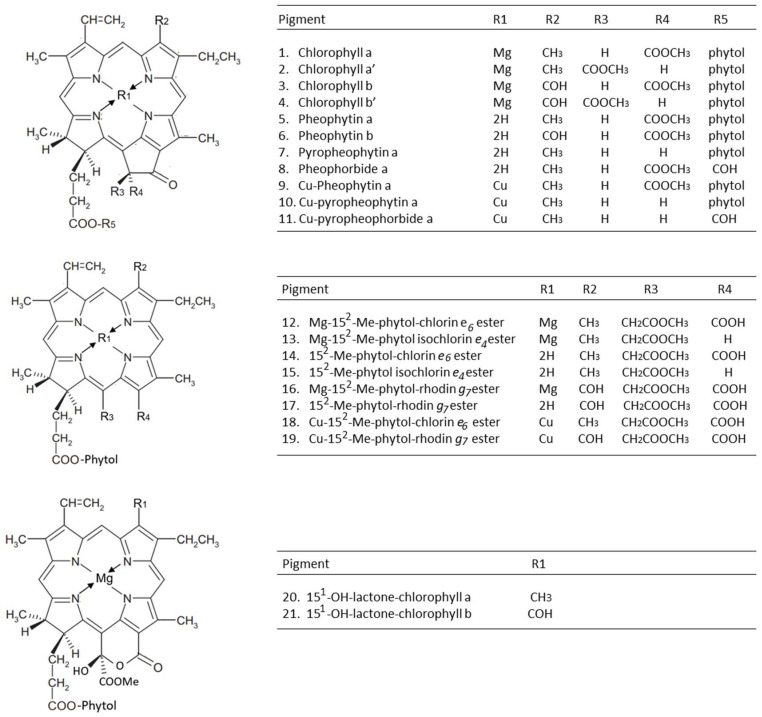
Chlorophylls (and their derivatives) found in alkali-treated table olives. From 1 to 8—[26]; 9 and 10—[8]; from 11 to 19—[5]; 20 and 21—[24].

As mentioned above, in Spanish-type olives, a short (6–8 h) treatment with alkali is followed by fermentation in NaCl brines, during which acids (mainly lactic acid) produced by microorganisms following sugar fermentation lower the pH, leading to the substitution of Mg^2+^ with 2 H^+^ in residual undephytylated chlorophylls and chlorophyllides, yielding pheophytins and pheophorbides, respectively [20]. Under such conditions, chlorophyll pigments may undergo Cu-chelation, a reaction that can have different consequences on the final coloration of drupes (see below) and occurs especially for pheophorbides a and b and 15^1^-OH-lactone-pheophytins a and b [7] and, to a lesser extent, for pheophytin b due to its lower nucleophilic capacity [27]. In contrast, the structural modification that results from the oxidation of the isocyclic ring for 15-glyoxylic acid pheophytin b favors such a reaction, as demonstrated by the detection of the corresponding copper complex. Compared to pheophytins, the absence of the alcohol phytol in pheophorbides theoretically increases ion interchanges because of a lower steric hindrance. However, the limited formation of the corresponding Cu complex has been reported, possibly because of the very low concentration of pheophorbides in the fruits [7].

Even if the entire process is shorter, Castelvetrano olives are maintained in high alkaline conditions for a longer period (2–4 weeks) compared to Spanish-type olives (6–8 h). Such conditions prevent the formation of Cu–chlorophyll complexes and prolong the solvolysis of the isocyclic ring [22]; in fact, the formation of such complexes requires the substitution of Mg^2+^ with 2H^+^ because Mg^−^ derivatives chlorophylls are not directly susceptible to Cu replacement [20]. Therefore, in such olives, the chlorophylls’ Mg is maintained but the chlorophylls are deeply allomerized and solvolyted [28]. Berlanga-Del Pozo investigated the effect of different NaOH concentrations and/or treatment times [29]. They found a direct relationship between the intensity of the alkali treatment and the degree of chlorophyll degradation and a higher susceptibility to oxidation and degradation for compounds of series b after one year of storage under refrigerated conditions.

When analyzing samples conserved using different chemical (acidified or non-acidified brine) and physical (pasteurized, sterilized, refrigerated, or RT) conditions, in their studies, Gandul-Rojas and Gallardo-Guerrero found that, at 24 months, most of the Castelvetrano chlorophylls are pheophytinized Mg-free derivatives (13^2^-OH-, 15^1^-OH-lactone and chlorin- and rhodin-type allomerized derivatives with structures for series a and b), achieved independently from acidification or pasteurization or sterilization [28]. On the other hand, in refrigerated and non-acidified samples, chlorophyll a remained the major pigment, and the transformation regarding the isocyclic ring of the chlorophyll structure experienced an increase in 15^2^-Me-phytol-isochlorin *e*_4_ ester content, along with an increase in pyropheophytin a and a small amount of Mg-free chlorophyll derivatives (15^2^-Me-phytol-chlorin e_6_ ester, pheophorbide a and pyropheophorbide a). Such olives also maintained the bright green color characteristic of Castelvetrano-style table olives, even though no Cu–chlorophyll complexes were detected in these samples [28]. 

Gandul-Rojas and Gallardo-Guerrero detected some Cu–chlorophyll complexes starting from 8 months of conservation in all samples (except the non-acidified and refrigerated samples). Up to ten different Cu–chlorophyll complexes were detected in olives packed and stored for 24 months, representing almost 7.5% of total chlorophyll pigments, an amount that, as stated by the authors, was not sufficient to shift the color of the olives to a bright green. The authors hypothesize that a uniform color change to bright green would require a greater than 11% portion of Cu–chlorophyll complexes [28]. The authors concluded that the higher proportion of Cu–chlorophyll derivatives in commercial green table olives processed through the Castelvetrano method is due to the deliberate addition of copper salts during processing or conservation and suggest quantitatively determining copper chlorophyll complexes in drupes presenting a doubtfully bright green color [28]. Nevertheless, high amounts of Cu–chlorophyll complexes have been reported in bright-green Castelvetrano olives, namely, Cu-15^2^-Me-phytol-chlorin e_6_ ester (26–48%), Cu-15^2^-Me-phytol-rhodin g_7_ ester (2–18%), Cu-pheophytin a (3–18%), and Cu-pyropheophytin a (1–5%) [5]. We recently reported that large amounts of copper Cu-derivatives of chlorophylls are produced even in olives stored in an alkaline brine (pH = 10) and that, on the other hand, no Cu-derivatives are formed when the concentration of copper in the drupes is very low (<3 mg/kg) [6].

## 4. Copper Derivatives and Color

The color shift of drupes from green to yellowish, induced by the pheophytinization reaction, negatively affects the perceptions of consumers because, usually, consumers associate a green color with naturalness. Due to the high pH of their conservation brines, Castelvetrano-style olives are particularly affected because of the thermal treatments necessary to ensure long-term preservation. In fact, unfortunately, both pasteurization and sterilization induce unwanted color changes. In particular, pasteurization provokes a rapid color shift to yellow due to pH conditions being below 4.3, while sterilization only induces color modifications during the subsequent storage period [3]. 

In order to ensure a green permanent color in the final product, a very commonly used but fraudulent manufacturing practice consists of the addition of food colorants [28,30,31]. Additionally, a patent for coloring green olives by sinking drupes with mixtures of chlorophyll chlorophyllin salts and copper chlorophyllin salts in temporary alkaline conditions for 1–72 h is available [32]. E-141, which is structurally related to chlorophyll and available both as liposoluble (E-141i) and hydrophilic (E-141ii) molecules, is the preferred colorant for olives, but it is prohibited in both of the above forms by the US Food and Drug Administration and by European Union regulations regarding table olives [11]. Due to its hydrophilic character and its high green color stability, E-141ii is the preferred colorant, even if the colorant may exhibit different compositions and different greening capabilities depending on the starting materials and manufacturing technologies. In fact, commercial-grade E-141ii is a mixture of numerous Cu-chlorin-type compounds derived from natural chlorophyll via saponification and Cu insertion. The chlorin-type structure is characterized by the presence of carboxylic groups in the chlorophyll molecule and by the absence of the ester-phytol at C-17^3^ and the isocyclic ring at C-15. Mortensen and Geppel reported that the major components of commercial E-141ii preparations are Cu chlorin *e*_6_, Cu chlorin *p*_6_, and Cu isochlorin *e*_4_ [33]. The issue with E-141ii usage is that it contains the same Cu–chlorophyll complexes produced during the conservation of the Castelvetrano-style table olives, making the discovery of fraudulence difficult when inappropriate analytical methods are applied. 

In fact, Gandul-Rojas and colleagues found that the spectroscopic characteristics of the main colorant peaks present in the E-141ii-adulterated Spanish or Castelvetrano olives were very similar to those of the chlorophyll Cu-derivatives spontaneously formed in green table olives [3]. Nonetheless, their solubility properties were different. In fact, the chromatograms obtained from analyzing the E-141ii colorant samples were significantly different from those of the naturally formed Cu–chlorophylls complexes [3]. Later, Negro et al. reported the presence of Cu compounds related to the E141ii colorant in 8 out of 16 samples of commercial table olives from the Italian market, concluding that the fraudulent addition of colorants to table olives is a quite common practice [30].

Finally, it is worth mentioning that in order to cope with the prohibition of the use of E-141, table olive regreening is often achieved by the prohibited addition of Copper (II) salts (which causes the formation of Cu–chlorophyll derivatives) at some point during post-harvest processing or conservation. This procedure takes advantage of the fact that the use of copper as an agricultural fungicide on plants with a maximum residue limit of 30 mg/kg is allowed [31], and it is not possible to analytically determine whether the copper present in a drupe is absorbed during the agricultural treatment of the plant or in a post-harvest bath. On the other hand, in our previous study, we reported how a clear greening effect can be obtained through treatments with copper, keeping the residues below the limits set by the legislation [6]. Gandul-Rojas and Gallardo-Guerrero, studying the pigment changes in table olives processed via the Castelvetrano method, stated that the highest proportion of Cu–chlorophyll derivatives found in those olives formed due to the deliberate addition of Cu (II) salts during debittering and/or conservation brines [28]. The authors suggest that the quantitative determination of copper–chlorophyll complexes in commercial samples of table olives should be necessary to detect such bad processing practices in table olives with a doubtfully bright-green color. Accordingly, Harp et al. found similar Cu derivatives, namely, Cu isochlorin *e*_4_ and Cu-15^2^-Me-chlorin *e*_6_, in commercial pale-yellow Castelvetrano olives treated either with copper salt (boiling 1 kg of olives for 30 min in 2 L of aqueous 50–250 mg/L CuSO4) or with E141ii (leaving 500 g of olives overnight at 20 °C in 1 L of 1000 mg/kg E141ii solution) [34]. As the amount between the two derivatives differed based on the treatment used, they proposed the use of their ratio to discriminate between table olives regreened via processing with copper salts or E141ii [34]. Other Cu derivatives, namely, Cu chlorin *e*_6_ and Cu pyropheophorbide a, were identified by Negro et al. in olive samples stirred for 3 h in already 1% CuSO4 and then stored in brine solutions for 48 h until pH stabilization to 5.4, in accordance with the fact that copper salt preferentially reacts at alkaline pH [30]. Finally, as mentioned above, after storing commercial-scale olive drupes at 4 °C in brines enriched with different amounts of copper salts, we recently demonstrated that a significant effect on both the content of Cu derivatives and on the green color of the drupes can be achieved artificially, even through using copper salts at concentrations below those allowed by legislation (<30 mg/kg) [6].

## 5. Green Staining

One particular consequence of the formation of copper derivatives during the debittering process is green staining. This phenomenon consists of the occasional appearance of brilliant green spots of different sizes distributed over the olive’s surface, contrasting with the naturally golden-yellow color of the fermented fruit. As mentioned in the introduction, the green staining alteration has been described only in table olives of the Gordal variety [35]. The phenomenon has been known for decades in olives processed according to the Spanish style, and its causes have been investigated since the 1990s [7,8,9,10].

The fact that the alteration is especially observed in Gordal olives has raised questions about some of the specific characteristics of this variety. Gordal olives are among the biggest olives employed for table olive production, and they show a high respiration rate during the post-harvest period [36]. Moreover, due to their relatively low phenolic compound content and particularly the bitter glucoside oleuropein, Gordal olives are sweet [37], and this characteristic favors lactic acid fermentation [38]. Chlorophyll transformation during the lactic fermentation of Gordal olives generates pheophytins, pheophorbides, and, unlike other olives, their respective pyroderivatives, the allomerized compounds 15-glyoxylic acid pheophytins a and b, 15-formylpheophytin a, and 15-hydroxylactone pheophytins a and b [10]. Gordal olives also possess higher polyphenol oxidase (PPO) activity than Hojiblanca but similar activity to Manzanilla, which are the two Spanish varieties most often employed for processing as table olives [37]. Because of its Cu-dependency, PPO was implicated in the formation of the Cu–chlorophyll complexes; however, a direct demonstration of this is still missing [39].

Gallardo-Guerrero and colleagues noticed that identical physical–chemical conditions can differently affect olives, possibly because of the differences in olive origins or stages of maturation [35]. This observation supports the idea that cell modification leading to the green staining alteration could also depend on the olives’ intrinsic characteristics. Accordingly, Sànchez and colleagues reported that the green staining alteration was completely absent in olives harvested in October, while the alteration was found in half of the barrels of olives harvested in September [12]. The authors had no explanation for the higher formation of green staining alterations in olives harvested early.

As demonstrated in electron microscopy studies, the formation of Cu-containing chlorophyll complexes requires cell rupture from the oxidative breakdown of the fruit chloroplasts [35]. The loss of the chlorophyll derivatives’ natural lipophilic environment [7] is necessary in order to have contact with the copper stored in pectin chains [40]. The involvement of cell integrity loss was demonstrated by the identification of pigment–lipoprotein degradation and by the reduced interaction between Cu–chlorophyll–protein complexes and thylakoid membranes [41].

Qualitatively, new Cu complexes are formed stepwise, and new metallo-chlorophyll compounds are detected as the fruits become more affected. In addition, the concentration of Cu complexes progressively increases, and the alteration finally spreads over the surface of the processed olive and becomes visible [7]. However, as metallo-chlorophyll complexes are spread all over the fruit, including across green staining-free areas, Gallardo-Guerrero et al. hypothesized that the compounds alone are not sufficient for the alteration to manifest [7]. Recent studies indicate that exogenous Cu (often used in agronomic practices) is irrelevant for the green staining appearance [20], as the endogenous Cu contained in pectin chains is sufficient for inducing the alteration [40]. In particular, higher calcium pectate and lower protopectin and soluble pectin fractions were reported in fruits displaying green staining, while total pectin content was similar to that of unaffected fruits.

As the peptidic fraction of the pigment–lipoprotein complex remains constant during green staining, the involvement of proteins was also excluded. Nonetheless, differences have been found in the amount of phospholipids, with green staining spots containing 1.6-fold less phospholipids than green staining-free surfaces [41].

Aiming to identify methods for minimizing the emergence of the green staining alteration, Sanchez and colleagues recently incubated processed olives at different temperatures in the presence of potassium sorbate, sodium benzoate, and citric acid (used as preservatives) [12]. The preservatives had no inhibitory effects on green staining appearance when the olives were treated at normal summer temperatures. However, the authors found that the green staining manifestation was inhibited if the olives were incubated at 65 °C, but the effects of temperature were not subsequently linked to the inhibition of some enzymatic reactions. Importantly, when comparing industrial- and laboratory-level processing, different susceptibilities to green staining were found, including an absence of the alteration in olives debittered and fermented at the laboratory level, a low incidence of the alteration in drupes debittered at the laboratory and fermented in the factory, and the highest incidence of the alteration in olives debittered at the factory, regardless of where they were fermented. This fact is possibly due to temperature differences during alkaline treatment. In fact, huge differences in batch sizes (around 3 kg in the laboratory fermenters and 10,000 kg in the factory tanks) can determine different increases in temperature, including those dramatically higher in industrial tanks [42]. Based on their findings, Sanchez and colleagues proposed a test to predict green staining susceptibility by incubating samples of olives in fermentation brine at 45 °C and assessing, after 20–25 days, the percentage of the green staining alteration [12].

## 6. Conclusions

This review has summarized the recent findings on the color changes that can take place in table olives during processing (before they reach tables for human consumption) while also taking into account adulteration practices. Chemical and physical modifications taking place in two alkali debittering approaches, the Spanish (or Sevillian) and Castelvetrano (or Campo Real) methods, and alterations in colors have been described, while fraudulent attempts to prevent such alterations in order to suit consumers’ preferences have been considered.

All these debittering processes determine the production of many molecules, some of which, such as Cu–complexes and derivates, are at the root of green staining. Nevertheless, the specific mechanisms involved in green staining formation are still unknown.

This literature review shows that further studies are needed to (1) better understand the precise molecular mechanisms of the green staining alteration; (2) investigate the processes that determine the differential formation of Cu–chlorophyll complexes in different olive cultivars and/or in different stages of the debittering process and successive storage; and (3) develop suitable and cost-effective analytical methods that enable the distinction of naturally formed Cu–chlorophyll complexes from banned E-141 additives.

## Figures and Tables

**Figure 1 molecules-28-06673-f001:**
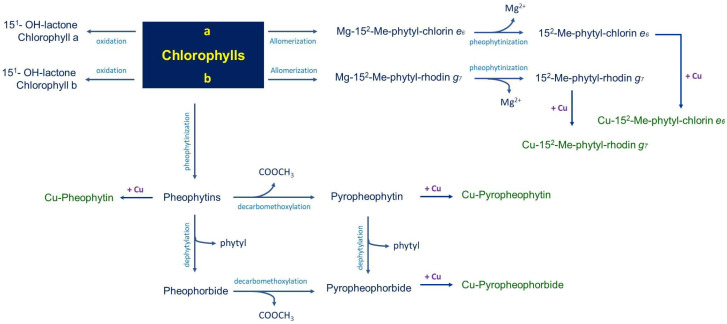
Chemical reactions that occur during olive chlorophyll degradation.

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
