# Peer review of "The Fate of Chlorophylls in Alkali-Treated Green Table Olives: A Review"

_molecules, 2023, doi:10.3390/molecules28186673_

Round 1

Reviewer 1 Report

molecules-2526797

The manuscript presents a succinct but clear review of the changes suffered by chlorophylls in green olives. The presentation is clear, and the language edition is rather correct. Only have a few suggestions.

-The title could be misleading. One alkali-treated olive style is outside the comments: the olives darkened by oxidation.  Then the title should be modified to properly define the products the manuscript refers to. Maybe, introducing green could be enough.

-Regarding table olive production, it is possible that the actual total amount of table olives consumed was above the figure the authors mentioned, but the official International Olive Council hardly exceeded 3 million tons. About or slightly below such production.

-According to the definition of “incubation”, this is not the most appropriate word to define the lye treatment. Immersed, submerged, etc. could be more adequate.

-“… a bright green the colour”. The sentence sounds rare. Additionally, maybe classifying such colours as bright is excessive since the presence of polyphenols always causes a slight browning even in more favourable conditions.  

L 38, “… natural dyes chlorophylls”. At least in the Spanish style, carotenes also play a sensible role, as the authors later mention.

-Please homogenise the in-text citation of authors, sometimes use just names and other family names. Family names are the usual in-text citations. 

Please, revise the manuscript again. Some suggestions are included in the comments.

Author Response

Dear Reviewer, please find below the point-by-point response to your comments. 

1) The title could be misleading. One alkali-treated olive style is outside the comments: the olives darkened by oxidation.  Then the title should be modified to properly define the products the manuscript refers to. Maybe, introducing green could be enough.

We agree with the Reviewer and included the word "green" in the title.

2) Regarding table olive production, it is possible that the actual total amount of table olives consumed was above the figure the authors mentioned, but the official International Olive Council hardly exceeded 3 million tons. About or slightly below such production.

We thank the Reviewer for the correction, "exceeding" has been substituted with "almost reaching" (line 23).

3) According to the definition of “incubation”, this is not the most appropriate word to define the lye treatment. Immersed, submerged, etc. could be more adequate.

We agree: "incubation" has been substituted with "immersion" (lines 28-30 and line 100).

4) “… a bright green the colour”. The sentence sounds rare. Additionally, maybe classifying such colours as bright is excessive since the presence of polyphenols always causes a slight browning even in more favourable conditions.  

The typo has been corrected (line 38). With respect to the colour classification, several publications indicate that the green colour can actually turn to bright, e.g. the references number 5 and 6 of the ms.

5) L 38, “… natural dyes chlorophylls”. At least in the Spanish style, carotenes also play a sensible role, as the authors later mention.

We agree, the sentence has been corrected to "olive’s natural dyes, including chlorophylls" (line 40).

6) -Please homogenise the in-text citation of authors, sometimes use just names and other family names. Family names are the usual in-text citations. 

Done throughout the text. 

Reviewer 2 Report

Ambra et al. presented an interesting literature-based study entitled "The fate of chlorophylls in alkali-treated table olives: a review". In this review, the author discusses the color changes of such molecules during alkali olive processing. 

The author provides an interactive representation of the chemistry of these pigments, which provides a unique perspective.

There are some minor comments.

1. Is there only a NaOH-based method available to remove the oleuropein, OR are there other methods available? Please state clearly.

2. Please add briefly about the chlorophyll pheophytinization reactions before introducing them on page 1, line 40.

3. Author stated, "the hydrolysis of bitter molecules promotes chlorophyll pheophytinization reactions, i.e. the substitution of Mg2+ by 2H+ in the chlorophyll chromophore group, that moves drupes color to yellow tones".

Are authors indicating the hydrolysis of oleuropein (or other bitter molecules or similar molecules)? Please clearly state. And how do these reactions change the color, please provide a generalized reaction that could simplify the concept and interest a broad audience.

4. "Other important chemical reactions that can influence in an opposite fashion drupes color are regreening phenomena associated to the insertion of endogenous or exogenous metals in the olives (Zn2+ or Cu2+) within the porphyrin ring of chlorophyll."

Does the regreening phenomenon have relevance in terms of greener processes or green chemical processing or recoloring? Please clarify.

5.  Author stated that such metal complexes with Zn and Cu provide stability and heat resistance to these compounds due to increased metal–porphyrin bond energies.

Are there other metals (divalent metal ions) than Zn and Cu that plays the same role? Please explain briefly how enhanced metal–porphyrin bond energies improve heat resistance. 

Author Response

Dear Reviewer, please find below the point-by-point response to your comments. 

1) Is there only a NaOH-based method available to remove the oleuropein, OR are there other methods available? Please state clearly.

Spontaneous fermentation without any chemical debittering treatment is also available, as in Greek-style black olives. This technical aspect has been mentioned more clearly (line 27). 

2) Please add briefly about the chlorophyll pheophytinization reactions before introducing them on page 1, line 40.

A brief description has been included, together with a reference for a broader audience (lines 43-47).

3) Author stated, "the hydrolysis of bitter molecules promotes chlorophyll pheophytinization reactions, i.e. the substitution of Mg2+ by 2H+ in the chlorophyll chromophore group, that moves drupes color to yellow tones".

Are authors indicating the hydrolysis of oleuropein (or other bitter molecules or similar molecules)? Please clearly state. And how do these reactions change the color, please provide a generalized reaction that could simplify the concept and interest a broad audience.

No, obviously we did not mean that hydrolysis of oleuropein promotes pheophytinization. We meant that the pH changes induced during oleuropein hydrolyzation promote chlorophylls pheophytinization. However, as the sentence could be misunderstood, it has been reformulated (line 42).

4) "Other important chemical reactions that can influence in an opposite fashion drupes color are regreening phenomena associated to the insertion of endogenous or exogenous metals in the olives (Zn2+ or Cu2+) within the porphyrin ring of chlorophyll."

Does the regreening phenomenon have relevance in terms of greener processes or green chemical processing or recoloring? Please clarify.

The term regreening if often used to refer to the phenomenon in which the olives, after having lost their green colour due to pheophytinization reactions, return to being green due to the inclusion of metals (mainly copper and zinc) in the porphyrin ring. The sentence has been made clearer (line 48-50).

5)  Author stated that such metal complexes with Zn and Cu provide stability and heat resistance to these compounds due to increased metal–porphyrin bond energies.

Are there other metals (divalent metal ions) than Zn and Cu that plays the same role? Please explain briefly how enhanced metal–porphyrin bond energies improve heat resistance.

Besides Cu and Zn, there are other metals that can substitute magnesium in chlorophylls, for example the three major toxic pollutants Cd2+, Hg2+ and Pb2+. In the review we refer only to copper as it is the only metal that is fraudulently used in the green olives processing. As regards to the heat resistance improvement of the metal–porphyrin bond energies, they are actually out of our expertise in food technology. Thus, the specific point related to heat has been deleted from the text and the bibliographic reference has been updated consequently (line 52-53).